# Hypoxia-Inducible Factor Prolyl Hydroxylase Inhibitors as a New Treatment Option for Anemia in Chronic Kidney Disease

**DOI:** 10.3390/biomedicines12081884

**Published:** 2024-08-18

**Authors:** Piotr Bartnicki

**Affiliations:** Department of Nephrology, Hypertension and Family Medicine, Medical University of Lodz, 90-549 Lodz, Poland; piotr.bartnicki@office365.umed.pl

**Keywords:** chronic kidney disease, anemia, erythropoiesis-stimulating agents, hypoxia-inducible factor system, prolyl hydroxylase inhibitors

## Abstract

Anemia plays an important role in chronic kidney disease (CKD) progression because it worsens the quality of life and increases the risk of cardiovascular complications in CKD patients. In such cases, anemia is mainly caused by endogenous erythropoietin (EPO) and iron deficiencies. Therefore, KDIGO and ERBP guidelines for anemia treatment in CKD patients focus on recombinant EPO and iron supplementation. A recent new treatment option for anemia in CKD patients involves blocking the hypoxia-inducible factor (HIF) system with prolyl hydroxylase inhibitors (PHIs), what causes increasing endogenous EPO production and optimizing the use of iron. Clinical studies have shown that the hypoxia-inducible factor prolyl hydroxylase inhibitors (HIF-PHIs) covered in this manuscript—roxadustat, vadadustat, daprodustat, and molidustat—effectively increase hemoglobin (Hb) levels in both non-dialyzed and dialyzed CKD patients. Moreover, these medicines reduce blood lipid levels and do not accelerate CKD progression. However, blockage of the HIF system by HIF-PHIs may be associated with adverse effects such as cardiovascular complications, tumorogenesis, hyperkalemia. and retinopathy. More extensive and long-term clinical trials of HIF-PHIs-based anemia treatment in CKD patients are needed, and their results will indicate whether HIF-PHIs represent an effective and safe alternative to EPO and iron supplementation for anemia treatment in CKD patients.

## 1. Introduction

Chronic kidney disease (CKD) is a significant health problem worldwide. It is estimated that about 10–15% of the population has CKD at various stages [1,2]. As CKD progresses to end-stage renal disease (ESRD), when patients require renal replacement therapy (RRT), the quality of life (QoL) of these patients deteriorates and the risk of death from cardiovascular causes increases [3,4]. Anemia plays an important role in the progression of CKD, QoL worsening, and increasing the risk of death from cardiovascular causes [5,6]. The incidence of anemia in CKD patients increases as glomerular filtration rate (GFR) decreases, which causes anemia in about 60% of non-dialysis-dependent CKD (NDD-CKD) patients and in more than 90% of patients treated with dialysis, i.e., dialysis-dependent CKD (DD-CKD) patients [7,8]. The prevalence of anemia in CKD in more recent references from Japan is estimated at 40.1% in stage G4 and 60.3% in stage G5 [9]; anemia prevalence (Hb < 10 g/dL) in a cohort of 5 million patients in the US with GFR < 74 mL/min is estimated at 59% in men and 71.4% in women [10]. The causes of anemia in CKD patients are complex [11]. The predominant role is played by a deficiency of endogenous erythropoietin (EPO) resulting from a progressive decrease in the number of active nephrons: under normal conditions, about 85% of EPO is produced by the perinephric cells of the renal tubules [12]. In CKD, important causative factors of anemia are absolute iron deficiency, resulting from blood loss or impaired iron absorption, and functional iron deficiency, resulting from the impaired release of iron from its stores following elevated hepcidin values [13]. It is also associated with chronic inflammation, comorbidities (e.g., diabetes, cancer), inhibition of the bone marrow response to EPO by uremic toxins, and vitamin B_12_ and folic acid deficiencies [12,14]. The introduction of erythropoiesis-stimulating agents (ESAs) and intravenous iron preparations into the treatment of anemia has definitely improved the outcome of treatment and QoL of these patients. It has been postulated that anemia treatment with EPO in NDD-CKD may contribute to slower CKD progression and reduce cardiovascular mortality [15,16]. On the other hand, ESAs and iron preparations-based therapy have some limitations, so it is necessary to justify developing new advanced approaches such as new medicines. Recently, a new anemia treatment option in CKD patients has opened up, with the introduction of prolyl hydroxylase inhibitors (PHIs) [17]. These medicines increase the production of endogenous EPO and optimize iron mobilization from its stores by influencing the hypoxia-inducible factor (HIF) system [18]. The purpose of this work is to present an overview of existing data on the anemia treatment in CKD patients and the current state of knowledge regarding the use of the hypoxia-inducible factor prolyl hydroxylase inhibitors (HIF-PHIs) (roxadustat, vadadustat, daprodustat, and molidustat) as a new treatment option in these patients.

## 2. Materials and Methods

A systematic review of the literature was conducted on the PubMed database of the National Library of Medicine, Bethesda, USA, to achieve the purpose of this manuscript. The keywords used in this search included: chronic kidney disease, anemia, erythropoiesis-stimulating agents, hypoxia-inducible factor system, prolyl hydroxylase inhibitors, roxadustat, vadadustat, daprodustat, and molidustat. Finally, 128 articles regarding the topics of interest were chosen, mainly randomized controlled trials (RCTs), meta-analyses, and systematic reviews. The majority of selected articles have been published in the last 5 years, some older, but important articles regarding CKD, anemia, HIF system, EPO, and guidelines for anemia treatment in CKD are included. Based on the analysis of the selected articles, a summary overview of the current knowledge on anemia treatment in CKD patients, including a new treatment option with HIF-PHIs, is provided in this paper.

## 3. Anemia Treatment in CKD

The introduction of human recombinant erythropoietin (rHuEPO) in the 1980s was a breakthrough in the treatment of anemia in patients with renal failure [19]. Initially, short-acting erythropoietin alpha (epoetin α) and beta (epoetin β) obtained from cell cultures by recombinant DNA technology were available [20]. Subsequently, EPOs with extended half-lives were introduced, making it possible to administer darbepoetin α (DA) every two weeks or methoxy polyethylene glycol-epoetin beta (MPG-EPO) every four weeks [21,22]. It seemed that continuous, slow stimulation of the EPO receptor with long-acting ESAs may be more beneficial for achieving and maintaining target hemoglobin (Hb) levels [23]. Also, continuous erythropoietin receptor activators (CERAs) such as MPG-EPO may have renal and cardioprotective effects [24]. Studies on the superiority of EPO types are divergent, with some indicating a higher mortality risk in patients treated with long-acting ESAs [25] and others indicating a high risk of CKD progression to ESRD and higher mortality following high doses of short-acting ESAs [26]. However, large meta-analyses on the efficacy and safety of ESAs indicate that there is currently no conclusive evidence for the superiority of any type of ESAs in the treatment of anemia in CKD patients [27,28]. The introduction of rHuEPO into the treatment of anemia has improved the QoL of CKD patients, reduced the need for blood transfusions, and reduced the clinical symptoms associated with anemia [29]. Despite this, clinical trials such as CHOIR, CREATE and TREAT have shown that overcorrection of Hb levels, especially above 13.5 g/dL, may result in an increased risk of cardiovascular events (myocardial infarction, stroke) including death, and may accelerate NDD-CKD progression to ESRD [30,31,32]. Optimal Hb concentration is currently believed to be between 10–12 g/dL, with the exact amount depending on risk factors, clinical status, and patient preferences. A key role in adequate erythropoiesis is played by iron. Most CKD patients have absolute or functional iron deficiency, which requires iron supplementation. Research suggests that intravenous iron is more effective at increasing Hb and serum ferritin (SF) levels and reducing the dose of ESAs in both NDD-CKD and DD-CKD patients. Oral iron is often poorly tolerated by patients, which limits the feasibility and efficacy of its use [33,34].

## 4. Guidelines of Anemia Treatment in CKD

Guidelines of anemia treatment in CKD patients given in two key documents—“Kidney Disease: Improving Global Outcomes” (KDIGO) and European Renal Best Practice (ERBP)—include the indications for EPO treatment, target Hb levels during EPO treatment, diagnosis of iron deficiency, a way of iron supplementation, control of SF, and transferrin saturation (TSAT) during EPO and iron treatment [35,36]. Although anemia treatment in CKD patients differs between countries and medical facilities [37], it is mainly performed according to KDIGO and ERBP guidelines. A summary of KDIGO and ERBP guidelines is presented in Table 1. However, these guidelines, dated 2009 and 2012, do not include more recent studies regarding the efficacy and safety of intravenous iron and a new treatment option with HIF-PHIs in CKD patients. Therefore, KDIGO decided to begin re-examination of 2012 guidelines and organized two controversies conferences to review new data from basic research and RCTs concerning anemia diagnosis and management in CKD patients. In 2019 the first conference focused on iron metabolism in CKD, including iron deficiency diagnosis, available therapies, and treatment targets [38]. The second conference was held in 2021 and focused on a new group of medicines, HIF-PHIs [39]. In 2024, a document was published by the ERBP board of the European Renal Association regarding the use of HIF-PHIs for anemia treatment in CKD [40]. This document presents the efficacy and safety of HIF-PHIs, discusses their place in anemia treatment in CKD patients according to the available evidence, and, finally, includes suggestions for clinical practice. These ERBP board suggestions for HIF-PHIs use in clinical practice are presented in Table 2.

## 5. Hypoxia-Inducible Factor (HIF) System

The HIF system induces an adaptive response to tissue hypoxia to prevent cell damage by improving oxygen delivery and reducing tissue oxygen consumption [41], which results in the production of EPO, mainly by interstitial perinephric cells in the kidney and by cells in the liver [42]. EPO binds to its receptor on the surface of erythroid progenitor cells in the bone marrow, improving the survival, maturation, and proliferation of red blood cells. The activity of the HIF system depends on the level of tissue oxygenation [43]. The HIF system consists of subunits α and β. Under hypoxia, HIF-1α accumulates and translocates to the cell nucleus where it attaches to HIF-β and induces the production of the heterodimer HIF-1αβ. HIF-1αβ stimulates the expression of various hypoxia-sensitive genes, including the gene for EPO, resulting in increased EPO production. Three isoforms of the HIF-α subunit are known as HIF-1α, HIF-2α, and HIF-3α, each of which can attach to the HIF-β subunit and induce the expression of various genes apart from the gene for EPO. Thus, the HIF-αβ heterodimer affects the expression of receptors for transferrin, vascular endothelial growth factor (VEGF), and endothelin-1 [44]. In addition, it has been postulated that the HIF system is involved in the control of cell metabolism and function [45], including immune cells [46], and affects total cholesterol and LDL fraction [47]. The HIF system is generally omnipresent: the transcription factor HIF-1α is produced in most cell types while HIF-2α is expressed in a more tissue-restricted manner [48,49]. mRNA expression of HIF-2α is mainly in the brain, heart, lung, kidney, pancreas, and intestine [50]. HIF-3α tissue expression has been reported in the heart, lungs, and kidneys [51]. Under normal tissue oxygenation, HIF-1α is degraded by hydroxylation with the participation of the enzyme prolyl hydroxylase (PH) and von Hippel-Lindau protein (pVHL); this prevents the fusion of HIF-1α and HIF-β, thus reducing the expression of the gene responsible for EPO production. Under hypoxia, PH is blocked and HIF-1α is not degraded, allowing for the formation of the HIF-1αβ heterodimer, which stimulates the gene responsible for EPO production [52]. The blockage of PH is the place of action for a new group of medicines HIF-PHIs, used for anemia treatment in CKD patients [53]. The HIF system is presented in Figure 1.

## 6. Hypoxia-Inducible Factor Prolyl Hydroxylase Inhibitors (HIF-PHIs)

HIF-PHIs increase endogenous EPO production by inhibiting PH [54]. Unlike ESAs, these medicines are administered orally and do not require special transport and storage conditions. This section presents meta-analyses of four HIF-PHIs: roxadustat, vadadustat, daprodustat, and molidustat.

### 6.1. Roxadustat (FG-4592)

Roxadustat has been approved for clinical use in Europe, Japan, and China. Clinical trials in patients with NDD-CKD treated with roxadustat have shown a greater increase in Hb levels compared with placebo [55]. Subsequent studies in patients on dialysis (DD-CKD) showed that roxadustat increased Hb levels independently of baseline iron balance, inflammatory markers (CRP), and RRT [56,57]. Roxadustat treatment yielded a greater increase in Hb in hemodialysis-treated ESRD compared with epoetin α [58]. In another study examining hepcidin, SF, and CRP levels in NDD-CKD patients, roxadustat treatment yielded a 16.9% reduction in hepcidin levels, a 30.9% reduction in SF levels, and a 15.3% increase in total iron-binding capacity (TIBC) compared with the control group [59]. An analysis of Phase 3 clinical trials comparing anemia outcomes in DD-CKD patients (the ROCKIES, PYRENEES, SIERRAS, and HIMALAYAS studies) found roxadustat to be superior in anemia correction compared with epoetin α. They also found that roxadustat treatment was less likely to entail blood transfusions than epoetin α [60,61,62,63]. Other Phase 3 studies (The ALPS, ANDES, and OLYMPUS studies) showed that roxadustat was more effective in achieving and maintaining target Hb levels compared with placebo in NDD-CKD patients [64,65,66]. The DOLOMITES study comparing the efficacy and safety of roxadustat with DA found that roxadustat did not have any advantages in achieving target Hb levels in the treatment of anemia in NDD-CKD patients [67]. An RCT from China regarding the efficacy and safety of roxadustat in patients with anemia on peritoneal dialysis (PD) showed that roxadustat more effectively corrected and maintained target Hb levels in comparison with ESA [68]. Investigators of this trial found that roxadustat decreased hepcidin levels and increased TIBC more effectively than ESA. An updated systematic review and meta-analysis of nine RCTs, which included 3175 patients in the roxadustat group and 2446 patients in the control group, showed that roxadustat effectively increased Hb levels and improved iron utilization parameters in NDD-CKD patients in comparison with the control group [69]. Next, an RCT, where 4277 patients with NDD-CKD and 3890 patients with DD-CKD were evaluated, showed that roxadustat effectively increased serum iron and TIBC and decreased hepcidin levels in comparison with the ESA group, with patients in the roxadustat group achieving target Hb levels with less intravenous iron supplementation in comparison with the ESA group [70]. Similar results regarding roxadustat efficacy for anemia treatment in CKD patients showed an updated meta-analysis of RCTs including 6518 patients [71]. ALTAI, a randomized, active-controlled, Phase 4 trial, investigated the efficacy of roxadustat versus ESA on gastrointestinal iron absorption in patients with anemia in stage 4/5 CKD and found no significant difference between roxadustat’s and ESA’s influence on iron absorption [72]. A key limitation of this study was recruitment difficulties and a small sample size.

### 6.2. Vadadustat (AKB-6548)

Vadadustat has been approved for clinical use in the US, Japan, and China. Phase 2a clinical trials in NDD-CKD showed that vadadustat caused increases in EPO levels comparable to physiological daily responses [73]. A multicenter Phase 2a study published by Pergola et al. found that vadadustat produced better anemia treatment results over placebo in achieving target Hb levels and a significant decrease in SF and hepcidin levels [74]. A clinical trial comparing the efficacy and safety of vadadustat with DA in patients with NDD-CKD and DD-CKD found that vadadustat did not have any clear advantage in maintaining target Hb values and that both study groups had similar safety profiles [75]. A post hoc analysis of the INNO_2_VATE clinical trial regarding DD-CKD patients receiving PD evaluated the safety and efficacy of vadadustat treatment in comparison with DA. The study showed that the safety and efficacy of vadadustat are not inferior to DA in this group of patients [76]. The authors concluded that vadadustat, an oral agent for anemia treatment for patients receiving peritoneal dialysis or home hemodialysis, is a better option than ESAs.

### 6.3. Daprodustat (GSK-1278863)

Daprodustat has been approved for clinical use in the US, Japan, and China. Clinical trials have found that daprodustat appeared to be well tolerated and to increase EPO levels in a dose-dependent manner [77]; treatment also appeared to increase Hb levels compared with placebo in NDD-CKD [78], maintain stable Hb levels after conversion from ESAs to daprodustat, and lower SF levels [79]. The ASCEND-NHQ randomized trial was conducted in 142 centers across 14 countries and evaluated the effects of daprodustat on Hb levels and the QoL in patients with NDD-CKD [80]. This study found that daprodustat treatment resulted in a significant increase in Hb level, a reduction in the need for blood transfusion, and a significant decrease in fatigue compared with placebo. Next, the randomized trial ASCEND-ID evaluated the efficiency and safety of daprodustat in anemia treatment in incident dialysis patients with CKD in comparison with DA and found no daprodustat advantage compared with DA [81]. The mean Hb level, intravenous iron use, and the need for blood transfusion during the evaluation were comparable in both groups. A meta-analysis of eight RCTs, where 8245 CKD patients were included, evaluated the safety and efficacy of daprodustat for anemia treatment in both DD-CKD and NDD-CKD [82]. This meta-analysis showed that daprodustat maintained the same efficacy in increasing Hb levels in both DD-CKD and NDD-CKD as ESAs; however, daprodustat significantly lowered hepcidin levels and increased TIBC in both groups in comparison with ESAs.

### 6.4. Molidustat (BAY-85-3934)

Molidustat has been approved for clinical use in Japan and China. Studies showed that molidustat increased the level of endogenous EPO [83] and reticulocyte count [84]. Effects of molidustat for anemia treatment in CKD were evaluated in the Phase 2 DIALOGUE study [85]. DIALOGUE 1 was a placebo-controlled study in NDD-CKD, DIALOGUE 2 was an open-label study where DA was switched to molidustat in NDD-CKD, and DIALOGUE 4 was an open-label study where epoetin α was switched to molidustat in DD-CKD. The results of this study showed that roxadustat increased Hb levels more effectively than placebo (increase in mean Hb of 1.4–2.0 g/dL) and darbepoetin α (increase in mean Hb of 0.6 g/dL), but no differences in mean Hb change between molidustat and epoetin α were confirmed. A randomized, open-label, Phase 3 study evaluated the efficacy and safety of molidustat for anemia treatment in NDD-CKD patients previously treated with DA [86]. This study found that daprodustat had no advantage over DA in maintaining Hb levels.

A meta-analysis including 13,146 patients evaluated the long-term efficacy and safety of HIF-PHIs for anemia treatment in CKD patients [87]. Thirty randomized controlled trials comparing treatment with HIF-PHIs (roxadustat, daprodustat, vadadustat, molidustat, desidustat, and enarodustat) versus ESAs or placebo were included in this meta-analysis. Investigators found that HIF-PHIs significantly increased Hb levels in comparison with the placebo or ESAs group. TIBC and transferrin levels were increased while hepcidin, SF, and iron levels were decreased in the HIF-PHI group in comparison with the placebo or ESAs group. The authors of this meta-analysis concluded that HIF-PHI treatment effectively increased Hb levels, promoted iron utilization, and was well tolerated for long-term use in CKD patients. They recommended to use HIF-PHIs in combination with iron supplementation for long-term anemia treatment in these patients. Further large clinical trials on the efficacy and safety of HIF-PHIs for anemia treatment in CKD patients are ongoing, and they also include the next HIF-PHIs, such as desidustat and enarodustat, which are not covered in this manuscript. The results of Phase 3 RCTs of anemia treatment with roxadustat, vadadustat, daprodustat, and molidustat are presented in Table 3 (NDD-CKD) and Table 4 (DD-CKD).

## 7. Comparison of the Effectiveness of Different HIF-PHIs

Generally speaking, data from the literature indicate that the efficacy of different HIF-PHIs in the anemia treatment of CKD patients is comparable, and all HIF-PHIs are equally effective as epoetin α and DA in anemia-correcting in these patients [92,93]. However, a network meta-analysis regarding a comparison of HIF-PHI treatment outcomes in anemia associated with CKD has shown some differences [94]. Results of 17 Phase 3 RCTs, evaluating roxadustat, vadadustat, and daprodustat, were analyzed in this meta-analysis. Three outcomes were evaluated: efficacy (Hb increasing), cardiovascular safety (time to the first major adverse cardiovascular event—MACE), and QoL. This meta-analysis included 7957 NDD-CKD and 12,320 DD-CKD patients. In NDD-CKD, roxadustat, vadadustat, and daprodustat were comparable in terms of efficacy and cardiovascular safety, whereas daprodustat was associated with a better benefit on QoL in comparison with roxadustat. In DD-CKD, roxadustat and daprodustat were associated with better efficacy than vadadustat, whereas the three HIF-PHIs were comparable in terms of cardiovascular safety. The next network meta-analysis regarding the efficacy of HIF-PHIs in DD-CKD patients involved 26 RCTs with 14,945 patients receiving one of five different HIF-PHIs: roxadustat, vadadustat, daprodustat, molidustat, and enarodustat [95]. This meta-analysis showed that roxadustat was the most effective HIF-PHI for Hb correction and that roxadustat and enarodustat were the most effective for reducing hepcidin and appropriate for patients with inflammation. On the other hand, the increased risk of hypertension and thrombosis associated with roxadustat was noted; therefore, in patients at risk for hypertension and thrombosis, the authors of the analysis propose molidustat or ESAs as the preferable treatment option.

## 8. Additional Actions of HIF-PHIs

HIF-PHIs exhibit other potentially beneficial actions in addition to their primary effects on anemia and iron metabolism. Phase 3 clinical trials showed that treatment with roxadustat reduced LDL cholesterol levels [96,97]. Similar cholesterol-lowering results were obtained for daprodustat, used for anemia treatment in the ASCEND-ND and ASCEND-D studies [88,90]. However, no cholesterol-lowering effect was observed for molidustat used for anemia treatment in CKD patients [98]. A meta-analysis of 25 RCTs including 17,204 participants confirmed that roxadustat and daprodustat were superior to epoetin α in lowering LDL cholesterol and total cholesterol levels, which are major risk factors for cardiovascular diseases among CKD patients; however, no data from recent studies have shown that this cholesterol-lowering effect of HIF-PHIs was associated with a lower incidence of MACEs [99]. Earlier clinical studies showed that anemia correction with HIF-PHIs did not substantially increase blood pressure [55,83,100]. On the other hand, there are new literature data indicating a higher risk of hypertension associated with HIF-PHI treatment, especially with roxadustat [69,95,101]. On this basis, it is reasonable to conclude that HIF-PHIs should be used with caution in patients with hypertension and they may be preferred in the treatment of anemia in CKD patients with hypotension. Apart from that, it is important that the literature data suggest that HIF-PHI treatment does not accelerate the progression of NDD-CKD to ESRD [102].

## 9. Potential Adverse Effects of HIF-PHIs

It should be considered that blocking the HIF system with HIF-PHIs may result in various adverse effects, particularly cardiovascular complications, tumorogenesis, hyperkalemia, and retinopathy.

### 9.1. Cardiovascular Complications

In CKD patients, anemia treatment with ESAs may be associated with an increased risk of cardiovascular events, especially when the Hb level exceeds 13.5 g/dL [103]. Similar relationships cannot be excluded with the use of HIF-PHIs. Four Phase 3 clinical trials (The PYRENEES, SIERRAS, HIMALAYAS, and ROCKIES studies) found that the rate of the first MACE for roxadustat, i.e.,1.09, was comparable with that for the placebo [104]; a similar value of 1.10 was achieved in three Phase 3 clinical trials in NDD-CKD (The ANDES, ALPS, and OLYMPUS studies) [105]. The ASCEND-D clinical trial examined the incidence rate of the first MACE in patients treated with daprodustat and ESA over a mean follow-up period of 2.5 years; the incidence was 25.2% in the daprodustat group and 26.7% in the ESA group [90]. In the ASCEND-ND study of NDD-CKD, this value was found to be 19.5% in the daprodustat group and 19.2% in the DA group over a mean follow-up period of 1.9 years [88]. On the other hand, a meta-analysis of eight clinical trials, where 8245 CKD patients were included, showed that daprodustat significantly reduced the incidence of the first MACE in DD-CKD in comparison with ESA [82]. The results of the INNO_2_VATE study, with the incidence of the first MACE in CKD patients treated with vadadustat and DA, showed that this rate in the group of DD-CKD patients was 18.2% in vadadustat group and 19.3% in DA group; in the NDD-CKD patients, the incidence was 22% for vadadustat and 19.9% for DA [91]. A post hoc regional analysis of the PRO_2_TECT study evaluated safety endpoints with vadadustat versus DA in NDD-CKD of ESA-naïve patients [89] and found no difference of the first MACE among the United States (US) patients and a higher risk among patients outside the US. The second part of this analysis evaluated NDD-CKD patients treated earlier with ESA and found that vadadustat was not inferior to DA in hematologic efficacy, but the risk of the first MACE was higher in the vadadustat group, especially in Europe [106]. A Phase 2 DIALOGUE study showed that molidustat treatment-emergent adverse events frequency was comparable to control groups, with no increased risk of cardiovascular complications found [85]. A systematic review and meta-analysis of 8806 patients evaluated cardiovascular and renal safety outcomes of roxadustat for anemia treatment in CKD patients and found that the risk of cardiac or kidney adverse events in the roxadustat group was not significantly higher in comparison with the placebo in both NDD-CKD and DD-CKD [101]. A pooled analysis of four Phase 3 RCTs showed that there was no evidence of increased risk of cardiovascular events or mortality in NDD-CKD and DD-CKD patients treated with roxadustat compared with ESA [107]. Moreover, in a multicenter, prospective, randomized trial, where 114 patients with DD-CKD were randomized to roxadustat and ESA group, the left ventricular mass index was evaluated after 12 months of treatment. The authors of this study found more regression of left ventricular hypertrophy in the roxadustat group in comparison with the ESA group on the maintenance of Hb levels of 10–12 g/dL [108]. On the other hand, an updated systematic review and meta-analysis of nine RCTs including 3175 patients in the roxadustat group and 2446 patients in the control group showed an increased incidence of serious adverse effects, especially deep vein thrombosis and hypertension in the roxadustat group [69]. A systematic review and meta-analysis of 12,821 dialysis patients evaluated the association of HIF-PHI treatment with cardiovascular events and death in comparison with ESAs and found no statistical difference among outcomes of the first MACE, myocardial infarct, stoke, or thrombosis events between the HIF-PHI and ESA group [109]. The next systematic review and meta-analysis of 15,144 patients evaluated cardiac and kidney adverse effects of HIF-PHIs for anemia in patients with CKD not receiving dialysis and found no significant difference in the risk of cardiac AEs between the HIF-PHI group and placebo [110]. These literature data indicate that HIF-PHIs (roxadustat, daprodustat, and vadadustat) and ESAs appear to have comparable cardiovascular safety profiles. They also show that the incidence of adverse cardiovascular events can reach 20% during the treatment of anemia in CKD regardless of whether patients are treated with ESAs or HIF-PHIs. On the other hand, RCTs and meta-analyses showed that long-term HIF-PHI treatment increased the risk of thrombotic events in comparison with ESAs [87,111]. For this reason, close monitoring of cardiovascular events is recommended before and during treatment with HIF-PHIs. A relevant warning about the increased risk of cardiovascular events can be found in the drug product information of HIF-PHIs approved for clinical use. Links to the drug information regarding the cardiovascular risk and other adverse effects are given below:

https://www.ema.europa.eu/en/medicines/human/EPAR/evrenzo (Accessed on 19 July 2024)

https://www.accessdata.fda.gov/drugsatfda_docs/label/2024/215192s000lbl.pdf (Accessed on 19 July 2024)

https://www.accessdata.fda.gov/drugsatfda_docs/label/2023/216951s000lbl.pdf (Accessed on 19 July 2024)

https://www.pmda.go.jp/files/000234811.pdf (Accessed on 19 July 2024)

### 9.2. Tumorogenesis

The literature raises the possibility that certain genes induced by the HIF system activation may be linked to cancer tumor formation and growth [112]. For example, in patients with von Hippel-Lindau (VHL) disease, the HIF system activation may result in the mutation in the pVHL causing VHL syndrome, characterized by tumors such as renal clear cell carcinoma, adrenal pheochromocytoma, pancreatic neuroendocrine tumor, and retinal and nervous system hemangioblastoma [113,114]. The pVHL as tumor suppressor is a subunit of a multiprotein ubiquitin ligase, which negatively regulates the expression of many hypoxia-inducible genes controlled by HIFs. The VHL ubiquitin ligase prevents inappropriate expression of these hypoxia-inducible genes when cells are grown in a large supply of oxygen by targeting HIFs for rapid ubiquitylation and degradation by the proteasome [115]. The next factor that may also result in tumorogenesis is the activation of the VEGF receptor by HIF-PHIs. Most clinical trials have not demonstrated any increased risk of cancer development during HIF-PHI treatment [116,117,118]; however, some studies indicate a higher incidence of malignancy in patients treated with daprodustat compared with DA [119] and in patients treated with molidustat compared with DA [86]. Data from the literature on clinical trials and animal models indicate a possible relationship between the HIF system and some cancers, such as colorectal, breast, and pancreatic cancer, osteosarcomas, and hematologic malignancies such as acute myeloid leukemia, acute lymphoblastic leukemia, and chronic myeloid leukemia [120]. Due to the possible risk of malignancy in patients treated with HIF-PHIs, careful and long-term tumor monitoring is recommended in this group of patients.

### 9.3. Hyperkalemia

Literature data regarding the incidence of hyperkalemia in patients treated with HIF-PHIs are inconclusive. Some authors indicate a higher incidence of hyperkalemia (the potassium serum concentrations > 5.5 mmol/L) in NDD-CKD patients treated with roxadustat and molidustat compared with the placebo and in DD-CKD patients treated with roxadustat compared with epoetin α [96,121]. Also, in clinical trials (The ALPS and OLYMPUS studies), a higher incidence of hyperkalemia was found in patients treated with roxadustat compared with the placebo [64,66]. In contrast, in other clinical trials (The HIMALAYAS and DOLOMITES studies), the authors report a higher incidence of hyperkalemia in ESA-treated patients compared with roxadustat [63,67]. Phase 3 clinical trials did not find a higher incidence of hyperkalemia in vadadustat-treated patients compared with ESA patients [122,123]. Since the literature data do not exclude the possibility of the occurrence of life-threatening hyperkalemia, it is recommended that blood potassium levels should be monitored before and during treatment with HIF-PHIs [124].

### 9.4. Retinopathy

As VEGF is known to stimulate retinal vascular formation and to be involved in the pathogenesis of diabetic retinopathy, degenerative changes in the retina, and macular edema, it cannot be excluded that treatment with HIF-PHIs may cause or exacerbate retinal changes. Phase 3 clinical trials on the use of roxadustat showed that retinal lesions, including retinal hemorrhages, occurred less frequently in patients treated with roxadustat compared with patients treated with DA [118,125]. Similar results were found in two Phase 3 clinical trials regarding the ophthalmologic effects of roxadustat in comparison with DA. They showed a lower incidence of a new or worsening retinal hemorrhage in patients treated with roxadustat (31.4%) compared with patients treated with DA (39.8%) [126]. The above studies showed the possibility of retinal hemorrhages in CKD patients treated with HIF-PHIs and ESAs; therefore, ophthalmologic examinations are recommended in these patients, especially when patients report visual disturbances [127].

In summary of potential adverse effects of HIF-PHIs, a systematic review and network meta-analysis of twenty trials with 14,947 participants was conducted to evaluate any adverse events (AEs) and serious adverse events (SAEs) of HIF-PHIs compared with ESAs. The authors of this review concluded that HIF-PHIs did not show significant differences from ESAs in terms of AEs and SAEs; they only observed statistical differences in gastrointestinal disorder, hypertension, and vascular access complications between different HIF-PHIs [128]. Other meta-analyses including 13,146 patients (30 randomized controlled trials), evaluated the long-term efficacy and safety of HIF-PHIs in anemia of CKD and showed that the risk of SAEs in the HIF-PHI group was comparable to the ESA group but increased compared with the placebo group. The most common adverse effects related to HIF-PHI treatment found in this meta-analysis were diarrhea, nausea, vomiting, peripheral edema, hyperkalemia, hypertension, and thrombosis events [87]. The most common side effects related to anemia treatment with HIF-PHIs in CKD patients are presented in Table 5.

## 10. Conclusions and Future Perspectives

The basis of anemia treatment in CKD patients, according to KDIGO and ERBP guidelines is still rHuEPO and iron supplementation. These guidelines do not include recent studies regarding using of intravenous iron and, especially, a new group of medicines HIF-PHIs. Therefore, the KDIGO controversies conference held in 2021 focused on the use of HIF-PHIs in anemia treatment in these patients. Based on the literature, it can be concluded that HIF-PHIs may become an important element of anemia treatment in CKD patients. Clinical studies have shown that this group of medicines effectively increases Hb levels in both NDD-CKD and DD-CKD patients. HIF-PHIs increase endogenous EPO production and improve iron mobilization and its availability for erythropoiesis. In addition, studies have shown that HIF-PHIs can reduce blood lipid levels and do not accelerate the progression of NDD-CKD to ESRD. HIF-PHIs also allow for an oral route of administration, which is a clear advantage in the treatment of anemia in NDD-CKD patients and those on PD or home dialysis. However, the blockage of the HIF system by HIF-PHIs may be associated with adverse effects such as cardiovascular complications, accelerated tumorogenesis, hyperkalemia, and retinopathy. On the other hand, some literature data about adverse effects of HIF-PHIs are conflicting. There are some RCTs where no advantage of HIF-PHIs versus ESAs in CKD anemia treatment was found. Many RCTs were conducted on a relatively small sample size and the follow-up was short. Also, a further direction is to conduct long-term clinical trials and prospective studies, including larger groups of participants to compare the efficacy and safety of HIF-PHIs and ESAs for anemia treatment in CKD patients. Their results should indicate whether HIF-PHIs are an effective and safe alternative to ESAs and iron supplementation in anemia treatment in CKD.

## Figures and Tables

**Figure 1 biomedicines-12-01884-f001:**
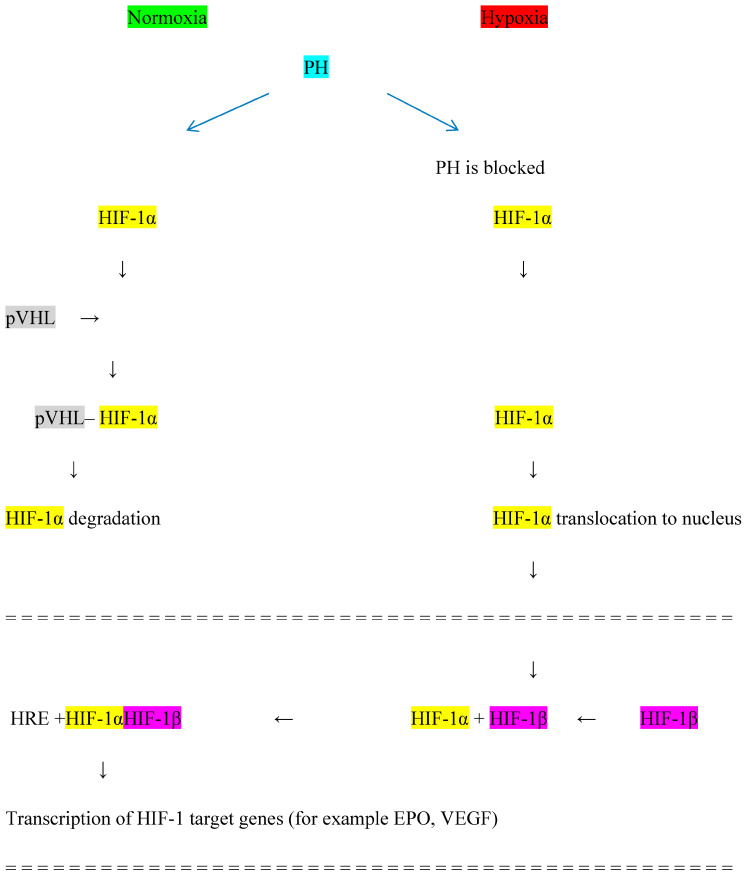
Hypoxia-inducible factor (HIF) system. PH—prolyl hydroxylase, pVHL—von Hippel−Lindau protein, HRE—hypoxia-responsive element, EPO—erythropoietin, VEGF—vascular endothelial growth factor.

**Table 1 biomedicines-12-01884-t001:** Summary of KDIGO and ERBP guidelines for anemia treatment in CKD [35,36].

	Iron Deficiency Diagnosis	Iron Supply	EPO Treatment Start	Hb Target	SF and TSAT under Iron Treatment
ERBP (2009)	SF < 100 ng/mL, TSAT < 20% if ESA naive; SF ≤ 300 ng/mL, TSAT ≤ 30% if ESA treated	Oral iron > 3 months (ND-CKD and mild-moderate anemia), iv iron (ND-CKD and severe anemia), or oral iron ineffective	Hb < 10 g/dL; Avoid Hb > 12 g/dL	10–12 g/dL; high-risk patients Hb around 10 g/dL	Avoid SF > 500 ng/mL and TSAT > 30%
KDIGO (2012)	SF ≤ 100 ng/mL, TSAT ≤ 20%	ND-CKD: a trial of iv iron or a 1–3-month trial of oral iron; DD-CKD: preference for iv iron	ND-CKD: Hb < 10 g/dL; DD-CKD: Hb 9–10 g/dL Avoid Hb < 9 g/dL	Hb ≤ 11.5 g/dL; Higher Hb if QoL improves and the patient accepts risks; Avoid > 13 g/dL	Stop iron supplementations if SF > 500 ng/mL

iv—intravenous.

**Table 2 biomedicines-12-01884-t002:** ERBP board suggestions for clinical practice regarding HIF-PHIs in anemia treatment in CKD patients [40].

Consider the Use of HIF-PHIs
NDD-CKD and PD patients
- Patient preference for oral treatment - Challenges to starting or receiving ESAs - Challenges to administering iron therapy or when increased iron availability is desired - ESA hyporesponsiveness or intolerance - Chronic inflammatory states (CRP ≥ 3 mg/L)
Hemodialysis patients
- Patient preference for oral treatment - Home hemodialysis - Hypersensitivity or unavailability of iv iron - ESA hyporesponsiveness or intolerance - Chronic inflammatory states (CRP ≥ 3 mg/L)
Use with caution
- Vascular access with a high risk of thrombotic complication - Retinal disorders (ophthalmology follow-up) - Autoimmune diseases (excluded in some clinical trials) - History of cured malignancy or without recurrence for at least 5 years - Kidney transplant recipients (not enrolled in clinical trials)
Avoid or use with extreme caution
- Patient with CV or thrombotic event in the previous 3 months - History of malignancy in the last 5 years - Polycystic kidney disease - Untreated proliferative diabetic retinopathy, macular degeneration, and retinal vein occlusion - Idiopathic pulmonary arterial hypertension
Administration key points
- Ensure adequate iron stores prior to treatment (SF > 100 µg/L, TSAT > 20%) - Individualize dose to achieve and maintain target Hb levels of 10–12 g/dL
Monitoring key points
- Avoid rapid rises or overcorrection in Hb levels - Monitor Hb levels at least monthly until the target Hb level - Monitor potassium and liver function

**Table 3 biomedicines-12-01884-t003:** The results of Phase 3 RCTs of anemia treatment with HIF-PHIs (NDD-CKD).

Study Name, Reference	HIF-PHI	Comparator	Baseline Hb (g/dL)	Hb Increase (g/dL)	Iron iv (%)	Transfusions (%)
ALPS [64]	roxadustat	placebo	9.1	1.98 versus 0.4	5.4 versus 5.9	8 versus 16.7
ANDES [65]	roxadustat	placebo	9.1	2.02 versus 0.18	2.5 versus 4.9	5.6 versus 15.4
ASCEND-ND [88]	daprodustat	darbepoietin α	9.9	0.74 versus 0.66	11.7 versus 11.8	2.8 versus 13.5
PRO_2_TECT [89]	vadadustat	darbepoietin α	9.1	1.43 versus 1.38	2.5 versus 2.3	5.1 versus 4.4
OLYMPUS [66]	roxadustat	placebo	9.1	1.75 versus 0.4	4.3 versus 7.9	13 versus 23
DOLOMITES [67]	roxadustat	darbepoietin α	9,5	2.5 versus 2.3	NA	NA
DIALOGUE 1 [85], Phase 2 study	molidustat	placebo	9.5	1.8 versus 0.3	NA	NA
DIALOGUE 2 [85], Phase 2 study	molidustat	darbepoetin α	10.8 versus 10.9	0.6 versus 0.1	NA	NA

**Table 4 biomedicines-12-01884-t004:** The results of Phase 3 RCTs of anemia treatment with HIF-PHIs (DD-CKD).

Study Name, Reference	HIF-PHI	Comparator	Baseline Hb (g/dL)	Hb Increase (g/dL)	Iron iv (%)	Transfusions (%)
ROCKIES [60]	roxadustat	epoietin α	10.2 versus 10.3	0.77 versus 0.68	NA	9.8 versus 13.2
PYRENEES [61]	roxadustat	epoietin α or darbepoietin α	10.8	0.51 versus 0.29	25.2 versus 56	9.2 versus 12.9
SIERRAS [62]	roxadustat	epoietin α	10.3	0.39 versus 0.09	17.1 versus 37	12.5 versus 21.1
ASCEND-ID [81]	daprodustat	darbepoietin α	9.5	1.02 versus 1.12	NA	12 versus 14
HIMALAYAS [63]	roxadustat	epoietin α	8.4 versus 8.5	2.57 versus 1.27	58 versus 89	7.3 versus 6.4
ASCEND-D [90]	daprodustat	epoietin α	10.4	0.28 versus 0.1	NA	15.7 versus 18.3
INNO_2_VATE [91]	vadadustat	darbepoietin α	9.4 versus 9.2	1.26 versus 1.58	10.5 versus 4,8	5.1 versus 4.4
DIALOGUE 4 [85], Phase 2	molidustat	epoetin α	10.5 versus 10.6	−0.2 versus −0.1	NA	NA

**Table 5 biomedicines-12-01884-t005:** The most common side effects related to anemia treatment with HIF-PHIs in CKD patients.

HIF-PHI	Side Effects	Source
Roxadustat	The most common side effects of roxadustat (which may affect more than 1 in 10 people) are hypertension, vascular access thrombosis, diarrhea, peripheral edema (swelling especially of the ankles and feet), hyperkalemia, and nausea. The most common serious side effects (which may affect up to 1 in 10 people) are sepsis, hyperkalemia, hypertension, and deep vein thrombosis. Roxadustat must also not be used in women who are breastfeeding or during the third trimester of pregnancy. https://www.ema.europa.eu/en/medicines/human/EPAR/evrenzo (Accessed on 19 July 2024)	European Medicines Agency
Vadadustat	Increased risk of death, myocardial infarction (MI), stroke, venous thromboembolism, and thrombosis of vascular access Hepatotoxicity Hypertension Seizures Gastrointestinal erosion Serious adverse reactions in patients with anemia due to chronic kidney disease and not on dialysis Malignancy https://www.accessdata.fda.gov/drugsatfda_docs/label/2024/215192s000lbl.pdf (Accessed on 19 July 2024)	The US Food and Drug Administration
Daprodust	Increased risk of death, myocardial infarction (MI), stroke, venous thromboembolism, and thrombosis of vascular access Risk of hospitalization for heart failure: increased in patients with a history of heart failure. Hypertension: worsening hypertension, including a possibility of hypertensive crisis occurring. Monitor blood pressure. Adjust antihypertensive therapy as needed. Gastrointestinal erosion: gastric or esophageal erosions and gastrointestinal bleeding have been reported. Not indicated for treatment of anemia of CKD in patients who are not dialysis-dependent Malignancy: may have unfavorable effects on cancer growth https://www.accessdata.fda.gov/drugsatfda_docs/label/2023/216951s000lbl.pdf (Accessed on 19 July 2024)	The US Food and Drug Administration
Roxadustat	Thromboembolism Hypertension Hepatic dysfunction Malignant tumors Retinal hemorrhage https://www.pmda.go.jp/files/000234811.pdf (Accessed on 19 July 2024)	Pharmaceuticals and Medical Devices Agency of Japan

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
