# Peer review of "Hypoxia-Inducible Factor Prolyl Hydroxylase Inhibitors as a New Treatment Option for Anemia in Chronic Kidney Disease"

_biomedicines, 2024, doi:10.3390/biomedicines12081884_

Round 1

Reviewer 1 Report

Comments and Suggestions for Authors

The text by Piotr Bartnicki et al. it is very well written, it describes the possible treatments of anemia during renal failure and the possible complications related to the use of HIF-PHIs.

The authors should include a descriptive table of the results and toxicity encountered in the main works regarding roxadustat, daprodustat, vadadustat and nolidustat depending on patients undergoing dialysis or not.

Comments on the Quality of English Language

The English language is adequate

Author Response

The authors should include a descriptive table of the results and toxicity encountered in the main works regarding roxadustat, daprodustat, vadadustat and nolidustat depending on patients undergoing dialysis or not.

The results of phase 3 RCTs of anemia treatment with roxadustat, vadadustat, daprodustat and molidustat are presented in Table 3 (NDD-CKD) and Table 4  (DD-CKD), page 12-13.

The most common side effects related to anemia treatment with HIF-PHIs in CKD patients are presented in Table 5, page 19-20.

Reviewer 2 Report

Comments and Suggestions for Authors

Dear Author,

The Manuscript represents a comprehensive attempt to review advances in chronic kidney disease (CKD) anemia treatment achieved by introducing prolyl hydroxylase inhibitors (PHIs). Notwithstanding the potential of attracting professional and scientific attention in addition to your efforts in preparation, the concerns that appeared during the review did not allow the recommendation for acceptance in the current form. Please consider the following comments and suggestions for the revised version.

Abstract

- Please explain whether the quoted drugs got approval for clinical practice.

- When mentioning that the drugs did not affect blood pressure, please try to be more specific and explain whether you referred to hyper- or hypotension.

- For the adverse effects, please specify the cardiovascular complications and tumors.

Introduction

- Please update the anemia frequency data using more recent references. The current articles (references 7 and 8) were published ten years ago.

- The erythropoiesis-stimulating agents (ESAs) and intravenous iron preparations-based therapy limitations are necessary to justify developing advanced approaches such as PHIs. This aspect should be among the aims.

Materials and methods

- The matching between the Manuscript methodological scope and the PRISMA2020 statement: An updated guideline for reporting systematic reviews (available at https://www.equator-network.org/reporting-guidelines/prisma/) is questionable. Therefore, the section introduces challenges and seems unnecessary for the revised version.

Anemia treatment in CKD

- Line 8: This point seems more suitable for quoting the target hemoglobin levels from Lines 22‒3.

Hypoxia-inducible factor (HIF) system

- When explaining the HIF system functioning (Lines 1‒4), please clarify which tissue/organs harbor this system. This additional explanation is necessary because the current text mentions only the predominant sites for EPO synthesis.

- Von Hippel-Lindau protein has a plethora of (patho)biological functions. Nevertheless, their description is scarce. Please consider efforts to improve it.

Hypoxia-inducible factor prolyl hydroxylase inhibitors (HIF-PHIs)

- To be more precise, please rephrase the title into Hypoxia-inducible factor prolyl hydroxylase inhibitors (HIF-PHIs) in CKD anemia treatment because section 7 reports on the beneficial effects in other CKD-accompanying comorbidities like cardiovascular pathologies or dislipidemia.

- The section would be more systematic if divided into subsections about each HIF-PHI. Please consider efforts to avoid merging anemia-related positive effects with those related to other comorbidities accompanying CKD that belong to Section 7. Notwithstanding the suitability of providing the details about HIF-PHIs in the advanced clinical trials, it seems objective to summarize the achievements regarding desidustat and enarodustat, which were also part of the metanalysis presented in the section.

- There is an inconsistency in presenting the findings from Reference 67. The study was multicentric (USA and Europe), but without any participating centers from Japan, as mentioned in the text. The info is lacking about the therapeutic agent to which the study compared vadadustat efficiency. Besides resolving this issue, additional efforts seem necessary to check any similar inconsistencies throughout the Manuscript.

- In the description of Reference 78, the abbreviation DA remained unexplained.

- If attainable, the readers would appreciate comments about comparisons between HIF-PHIs regarding their efficiency.

Cardiovascular complications

- Please provide links to the drug information quoting the cardiovascular risk.

Hyperkalemia

- For less challenging interpretation, please quote the potassium serum concentrations used as the hyperkalemia cut-offs in different studies.

Highlights

- Shorter text would make it easier for the readers to remember and accept the key messages. 

Technical suggestions

- The language editing would additionally contribute to the overall quality. Currently, the text contains inconsistencies regarding conciseness and clarity.

- Please enumerate lines in the revised version.

- Additional efforts are necessary to ensure uniform abbreviation usage throughout the Manuscript. For example, the differences are present between the text and the tables. Additionally, some of the abbreviations from the tables lack explanations.

Comments on the Quality of English Language

The Comments to the Authors contain suggestion regarding the Quality of the English Language

Author Response

Abstract

Please explain whether the quoted drugs got approval for clinical practice. It is explained in the desciption of HIF-PHIs, pages 9,10,11.

When mentioning that the drugs did not affect blood pressure, please try to be more specific and explain whether you referred to hyper- or hypotension.Explanation is given on page 15, lines  2-7.

For the adverse effects, please specify the cardiovascular complications and tumors.Cardiovascular complications are given in chapter 9.1 (page 15-17) and Table 5 (page 19-20). Tumors in chapter 9.2, page 17, lines 15-18.

Introduction

Please update the anemia frequency data using more recent references. The current articles (references 7 and 8) were published ten years ago. Update data given on page 2, lines 10-13, two new references [9,10].

The erythropoiesis-stimulating agents (ESAs) and intravenous iron preparations-based therapy limitations are necessary to justify developing advanced approaches such as PHIs. This aspect should be among the aims. Tis aspect given on page 2, lines 25-27.

Materials and methods.

The matching between the Manuscript methodological scope and the PRISMA2020 statement: An updated guideline for reporting systematic reviews (available at https://www.equator-network.org/reporting-guidelines/prisma/) is questionable. Therefore, the section introduces challenges and seems unnecessary for the revised version. 

Review articles that can be found in Biomedicines often contain a material and methods section, where the selection of articles is not made in accordance with PRISMA 2020. According to the instructions for authors, such a chapter is not obligatory for review articles. The reviewer suggests not to include this chapter in the revised version. I left it but in a shortened version.

Anemia treatment in CKD

Line 8: This point seems more suitable for quoting the target hemoglobin levels from Lines 22‒3.In line 8 it is "target Hb" but in line 22-23 is "optimal Hb". I think it is ok and does not require change.

Hypoxia-inducible factor (HIF) system

When explaining the HIF system functioning (Lines 1‒4), please clarify which tissue/organs harbor this system. This additional explanation is necessary because the current text mentions only the predominant sites for EPO synthesis. Explanation is given on page 7, lines16-20.

Von Hippel-Lindau protein has a plethora of (patho)biological functions. Nevertheless, their description is scarce. Please consider efforts to improve it.Explanation given in chapter 9.2, page17, lines 6-10.

Hypoxia-inducible factor prolyl hydroxylase inhibitors (HIF-PHIs)

To be more precise, please rephrase the title into Hypoxia-inducible factor prolyl hydroxylase inhibitors (HIF-PHIs) in CKD anemia treatment because section 7 reports on the beneficial effects in other CKD-accompanying comorbidities like cardiovascular pathologies or dislipidemia. The title of manuscript is changed.

The section would be more systematic if divided into subsections about each HIF-PHI. Please consider efforts to avoid merging anemia-related positive effects with those related to other comorbidities accompanying CKD that belong to Section 7. Notwithstanding the suitability of providing the details about HIF-PHIs in the advanced clinical trials, it seems objective to summarize the achievements regarding desidustat and enarodustat, which were also part of the metanalysis presented in the section. Chapter 6 is divided into 4 parts: roxadustat, vadadustat, daprodustat and molidustat. Desidustat and enarodustat are not covered in my manuscript, information abot this are placed on page 1 (Abstract), page 3 (Introduction), page 6 (Chapter 60). It is placed information about desidudstat and enarodustat in the end of chapter 6, page 11-12: "Further large clinical trials on the efficacy and safety of HIF-PHIs for anemia treatment in CKD patients are ongoing, they also including the next HIF-PHIs such as desidustat and enarodustat, which are not covered in this manuscript".

There is an inconsistency in presenting the findings from Reference 67. The study was multicentric (USA and Europe), but without any participating centers from Japan, as mentioned in the text. The info is lacking about the therapeutic agent to which the study compared vadadustat efficiency. Besides resolving this issue, additional efforts seem necessary to check any similar inconsistencies throughout the Manuscript. Text conected with Reference 67 is corected: "A Japanese" is removed. It is mentioned that vadadustat was compared to DA, that is darbepoetin alfa: chapter 6.2, page 10, line 6.

In the description of Reference 78, the abbreviation DA remained unexplained. The fist explanation of abbreviation DA is placed earlier in text, chapter 3, page 3, line 5.

If attainable, the readers would appreciate comments about comparisons between HIF-PHIs regarding their efficiency. It is placed in the new chapter 7, page 14.

Cardiovascular complications.

Please provide links to the drug information quoting the cardiovascular risk. Are placed on page 17, lines 5-8 and in Table 5, pages 19-20.

Hyperkalemia.

For less challenging interpretation, please quote the potassium serum concentrations used as the hyperkalemia cut-offs in different studies. Given in chapter 9.3, page 18, lines 2-3.

Highlights

Shorter text would make it easier for the readers to remember and accept the key messages. Chapter has been shortened.

Technical suggestions

The language editing would additionally contribute to the overall quality. Currently, the text contains inconsistencies regarding conciseness and clarity.  

English language was re-checked and several corrections were made despite the first reviewer's lack of comments on the quality of the language.

Please enumerate lines in the revised version. Line numbers of corected text are given in explanations to reviewer's comments.

Additional efforts are necessary to ensure uniform abbreviation usage throughout the Manuscript. For example, the differences are present between the text and the tables. Additionally, some of the abbreviations from the tables lack explanations. 

Manuscript was checked several times for abbreviations. There were gaps and differences. They have been corrected.

Round 2

Reviewer 2 Report

Comments and Suggestions for Authors

Dear Authors,

The efforts resulted in a revised MAnuscript that met the criteria to recommend acceptance for publication. Please note that, in the Introduction, the abbreviation G4 lacks explanation and that the commas separated the decimals throughout the Manuscript. These issues necessitate corrections during the technical editing.